# Clonal architecture predicts clinical outcomes and drug sensitivity in acute myeloid leukemia

Brooks A. Benard [1,2], Logan B. Leak [2,3], Armon Azizi[1], Daniel Thomas [1,4], Andrew J. Gentles [5] & Ravindra Majeti [1✉]

The impact of clonal heterogeneity on disease behavior or drug response in acute myeloid leukemia remains poorly understood. Using a cohort of 2,829 patients, we identify features of clonality associated with clinical features and drug sensitivities. High variant allele frequency for 7 mutations (including *NRAS* and *TET2*) associate with dismal prognosis; elevated *GATA2* variant allele frequency correlates with better outcomes. Clinical features such as white blood cell count and blast percentage correlate with the subclonal abundance of mutations such as *TP53* and *IDH1*. Furthermore, patients with cohesin mutations occurring before *NPM1*, or transcription factor mutations occurring before splicing factor mutations, show shorter survival. Surprisingly, a branched pattern of clonal evolution is associated with superior clinical outcomes. Finally, several mutations (including *NRAS* and *IDH1*) predict drug sensitivity based on their subclonal abundance. Together, these results demonstrate the importance of assessing clonal heterogeneity with implications for prognosis and actionable biomarkers for therapy.

[1] Department of Medicine, Division of Hematology, Cancer Institute, Stanford University, Stanford, CA, USA. [2] Cancer Biology Program, Stanford University, Stanford, CA, USA. [3] Department of Biology, Stanford University, Stanford, CA, USA. [4] Adelaide Medical School, University of Adelaide, Adelaide, SA, Australia. [5] Department of Medicine (Biomedical Informatics/Quantitative Sciences unit), Stanford University, Stanford, CA, USA. ✉email: rmajeti@stanford.edu

Acute myeloid leukemia (AML) is an aggressive cancer that develops from the accumulation and clonal expansion of somatic driver mutations in hematopoietic stem and progenitor cells[1–3]. While the mutational burden in AML is relatively low[4,5], patients present with heterogeneous disease characterized by variable clonal and subclonal genotypes and a relatively stable cytogenetic karyotype[6–9]. These properties endow AML with several advantageous features for studying the clonal evolution of cancer. Recent studies have noted that subclones within leukemia samples can show differences in morphology and immunophenotype, response to chemotherapy or hypomethylating agents, proliferation in response to myeloid growth factors in vitro, and engraftment in immunocompromised mice[10–14]. Emerging reports have also suggested that increased mutation burden and clonal heterogeneity correlate with worse survival outcomes[8,15,16]. Whether the increased risk is associated with sequential accumulation of mutations in the same clone or distribution across independent subclones is unknown. There is also increasing evidence that clonal dynamics impact the development and clinical outcomes of AML. For example, the subclonal prevalence of specific mutations, as measured by variant allele frequency (VAF), has been shown to predict progression from clonal hematopoiesis of indeterminant potential to myeloid malignancy, in addition to predicting time-to-relapse[17–21]. Recent work has also correlated higher VAF of *ASXL1, DNMT3A, JAK2, TET2,* and *TP53* mutations with worse outcomes in AML patients with intermediate risk disease[22]. Additionally, *TP53* VAF and allelic state have recently been shown to be prognostic in myelodysplastic syndromes (MDS)[23–25]. Despite the prognostic utility of VAFs in these and other focused genotypes in AML[26,27], there has yet to be a comprehensive analysis using VAFs to assess risk stratification or identify differences in response to therapy.

Emerging single-cell DNA-sequencing studies have provided unprecedented resolution of leukemia evolution and clonal structure, largely recapitulating trends observed in bulk sequencing[28–30]. Unfortunately, due to cohort sizes, these studies are insufficiently powered to correlate granular features of clonality with clinical outcomes. In contrast, several large bulk sequencing studies of AML are powered to provide insights into broad clonal trends, correlation of combinatorial genetic features to clinical outcomes, and biomarkers of therapeutic response to targeted therapies[7–9,31,32]. However, clonal evolution and VAF have yet to be systematically integrated with response to therapy or with more granular risk stratification in these cohorts. To address these questions, we aggregated clinically annotated cohorts of genotyped AML patients and analyzed the clonal architecture of recurrent somatic mutations in order to identify potential correlations with features of disease presentation, survival outcomes, and drug sensitivity.

Here, we show unreported survival associations with mutation co-occurrence patterns, VAF, and clonal evolutionary trajectories that would not have been observed using traditional analyses (mutation present/absent). We also infer unique drug sensitivities based on the clonal abundance of specific mutations, providing insights as to how certain mutations may influence the bulk leukemia depending upon their subclonal abundance. Overall, our findings validate the clinical importance of incorporating clonal analysis into the molecular evaluation and treatment of AML.

## Results

**Cohort curation and summary**. A total of 13 studies[7–9,33–42] were aggregated into a database comprising of 2829 patient samples profiled with an admixture of DNA-sequencing modalities and ex vivo drug screening (Fig. 1a and Supplementary Fig. 1a). A summary of clinical characteristics and treatment histories for this cohort can be found in the Supplementary Material (Supplementary Fig. 1b–j). Study aggregation recapitulated previously reported mutation frequencies in AML, with broad similarities in mutation patterns of co-occurrence, mutual exclusivity, and prognosis (Fig. 1b)[8,31]. Notably, our augmented cohort size identified *FLT3-TKD* and *RAD21* as mutations previously unreported to be associated with favorable outcomes (Supplementary Fig. 1k). For studies reporting VAFs, bimodal distributions highlighted the presence of clonal and subclonal populations (Supplementary Fig. 1l). VAF distribution for individual genes also showed significant variability in mean VAF, reflecting different patterns of clonal and subclonal dynamics between mutations (Supplementary Fig. 1m). In total, 2038 de novo patients with VAF and survival outcomes data were identified for subsequent analyses.

**Distinct genotypes predict clinical presentation and survival**. AML is a cancer of low mutation-burden[4,5] and thus provides an ideal model disease to investigate how mutations interact to promote and drive disease progression. Specific mutation combinations can be selected for during cancer development and drive clonal expansion because their co-occurrence improves tumor fitness, thus driving epistatic patterns observed in cancer sequencing studies[43,44]. Theoretically, if epistasis is driven by the selective pressure of leukemia fitness, then there might be correlations between broad binary epistatic trends and features of disease presentation and patient outcomes. To better understand how the relationship between co-occurring mutations might influence aspects of disease, we first defined the associations between individual mutations and features of clinical presentation (Supplementary Fig. 2). Multiple associations were identified, for example the finding that *IDH2*, but not *IDH1*, mutations associated with lower lactate dehydrogenase (LDH) levels (Supplementary Fig. 2).

We next used our augmented cohort size to define the statistical co-occurrence and mutual exclusivity of frequent mutations (Fig. 2a). In addition to previously reported associations[8,9,31], we identified several unreported co-occurrence (e.g. *EZH2* and *CBL*; odds ratio = 4.7; $q = 0.046$) or mutually exclusivity patterns (e.g. *TP53* and *KRAS*; odds ratio = 0.07; $q = 0.017$) (Supplementary Fig. 3a). Next, we analyzed the correlation between pairwise mutations and survival (Fig. 2b). We identified multiple cases where co-occurring mutations either re-stratified (e.g., *NRAS* + *SRSF2*) or strengthened the trend (e.g., *NRAS* + *RAD21*) of survival associations seen at the single mutation level (compare Supplementary Figs. 1k and 3b). Interestingly, overlaying the epistatic landscape with survival associations showed no correlation between pairwise epistasis (odds ratios) and survival (hazard ratios) (Fig. 2c). More specifically, there was no enrichment for worse outcomes based on an increased odds ratio of co-occurrence (Chi-squared test; $p = 0.15$), suggesting that an increased frequency in mutation co-occurrence does not necessarily drive improved leukemia fitness. One possible explanation is that "co-occurring" mutations in the same leukemia might not occur in the same cells, but instead occupy distinct clonal/subclonal populations. For example, when we investigated the relationship between co-occurring *NRAS* and *KRAS* mutations, we observed a strong pattern of inverse clonality (high VAF in one gene associated with low VAF in the other), suggesting these mutations arise in independent cellular populations (Supplementary Fig. 3c). We observed a similar pattern between *NRAS* and *PTPN11* mutations (Supplementary Fig. 3d). Given their functional redundancy in RAS/ MAPK signaling, these results suggest mutual exclusivity in clones harboring hyperactivation of RAS/MAPK pathway

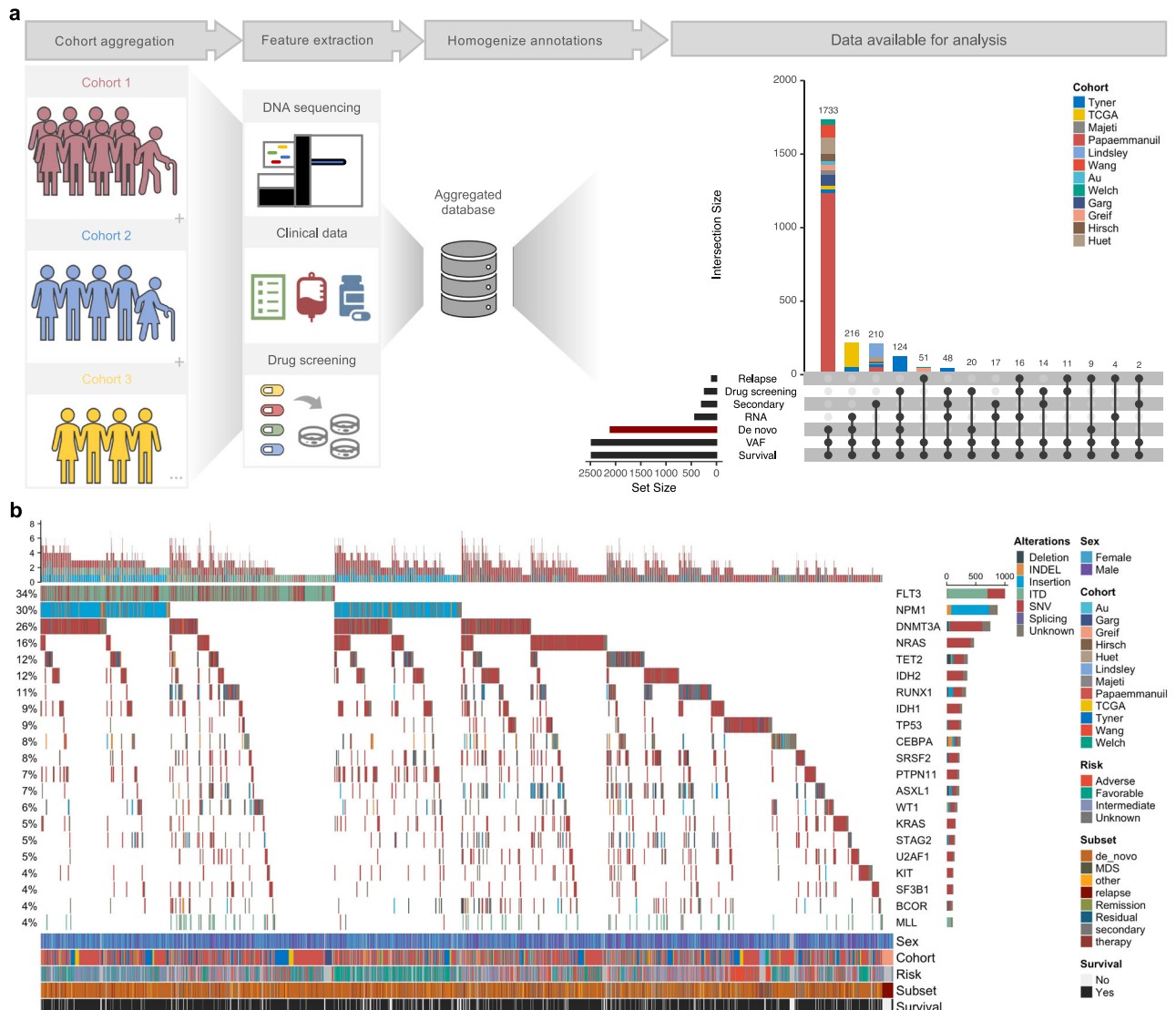

**Fig. 1 Cohort curation and summary. a** A systematic literature review was performed to identify studies reporting clinically annotated samples and/or molecularly profiled cohorts of adult AML. 12 cohorts met inclusion criteria and were curated for mutations, drug screening results, clinical features, and survival outcomes. These studies were then aggregated into a uniformly annotated database with an admixture of overlapping data types available for analysis. **b** Oncoprint for the most frequently mutated genes in our cohort. Each column is an individual sample ($n = 2914$) and the color of the vertical line represents the type of mutation reported. Sex, cohort, ELN 2017 risk group, subset, and survival information is indicated on the bottom of the plot. Source data are provided as a Source Data file.

signaling in AML. Indeed, recent single-cell genotyping of myeloid malignancies offers definitive evidence that mutations in RAS/RTK genes tend to be mutually exclusive[29,30].

We next tested the hypothesis that co-occurring mutations might drive differences in disease presentation. Pairwise mutation analysis revealed strong associations between several genotypes and the abundance of white blood cells (WBCs), platelets, LDH, and blast percent in the bone marrow and peripheral blood (Supplementary Fig. 3e). Of note, the effect size of mutations in *FLT3*, *IDH2*, *TP53*, *CEBPA*, and *NRAS* varied drastically depending on their co-occurring mutations (compare Supplementary Figs. 3e and 2), suggesting a complex interplay between clonal genotypes and features of disease presentation.

We also investigated if multiple mutations in the same gene or functional category/pathway were correlated with clinical features or risk. Compared to patients with only one mutation, multiple mutations in *CEBPA* predicted high platelet counts, lower hemoglobin counts and peripheral blood blast percentages, older

age, and better survival outcomes (Supplementary Fig. 3f–g). Multi-hit *TP53* correlated with higher bone marrow and peripheral blood blast percentages and decreased age while multiple mutations in *FLT3-TKD* was associated with older age (Supplementary Fig. 3f). Multiple mutations in genes related to transcription were associated with decreased WBC counts, hemoglobin levels, and peripheral blast percentages, while also associating with increased platelet counts and improved outcomes (Supplementary Fig. 3h, i). More than one mutation in tumor suppressors predicted higher bone marrow blast percentages, whereas multiple mutations in RTK/RAS signaling components correlated with improved outcomes (Supplementary Fig. 3h, i). Finally, multiple mutations in genes related to chromatin remodeling and cohesin components correlated with lower WBC counts, younger age, and worse survival outcomes (Supplementary Fig. 3h, i).

We next augmented our analysis to investigate the prognostic association of triple-mutated genotypes compared to pairwise mutated genotypes. In total, 48 distinct triple-mutated genotypes

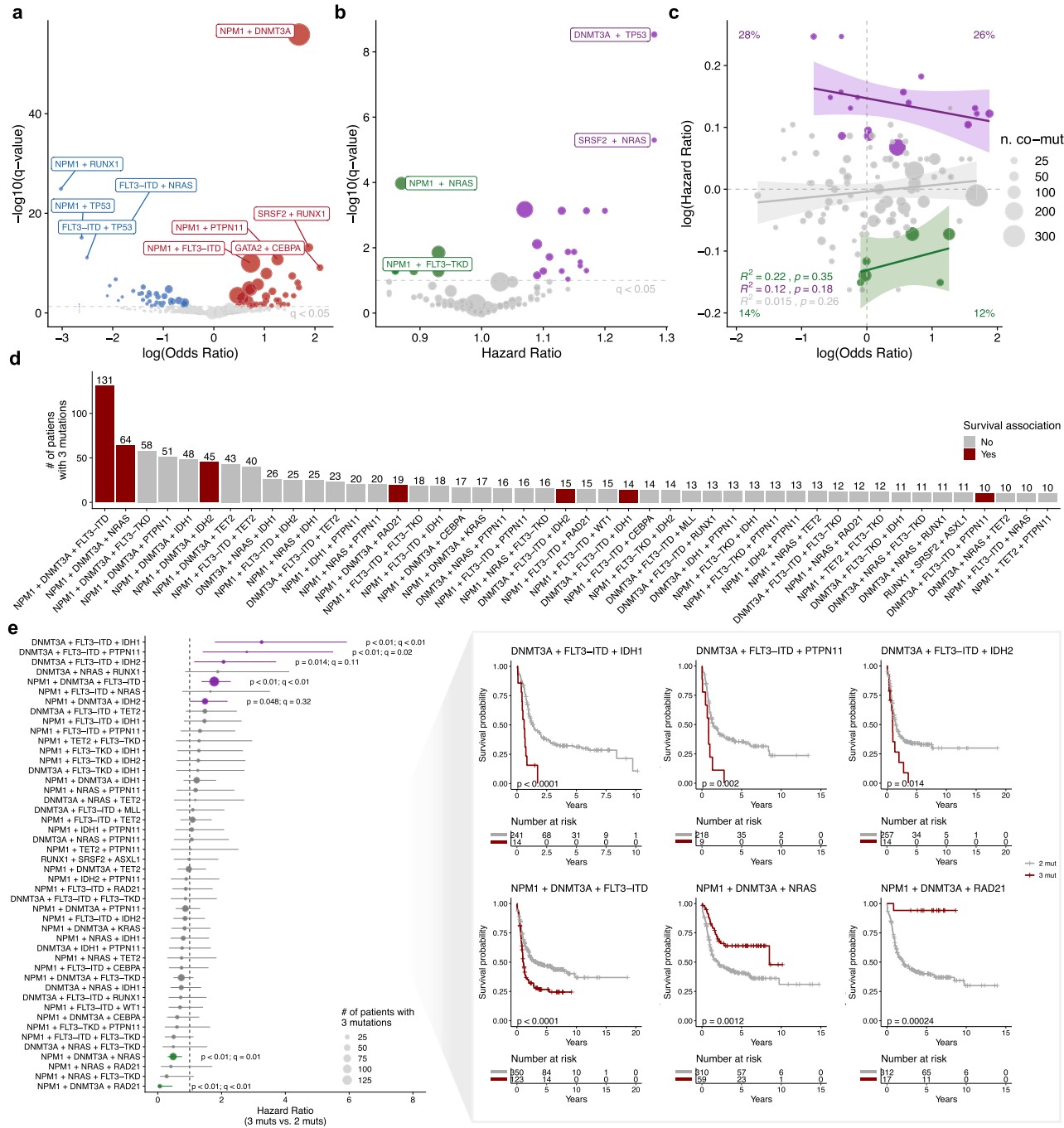

had at least ten patients with all three mutations present (Fig. 2d). We confirmed the association of worse outcomes in patients with mutations in *DNMT3A*, *NPM1*, and *FLT3-ITD* ($q < 0.01$; HR = 1.77; 95% CI: 1.37–2.29), in addition to improved outcomes in patients with mutations in *DNMT3A*, *NPM1*, and *NRAS* ($q = 0.01$; HR = 0.48; 95% CI: 0.31–0.76; Fig. 2e)[8]. Importantly, our analysis revealed unreported survival correlations for four genotypes: *DNMT3A:FLT3-ITD:IDH1* ($q < 0.01$; HR = 3.26; 95% CI:1.79–5.93), *DNMT3A:FLT3-ITD:IDH2* ($q = 0.11$; HR = 2.06; 95% CI: 1.14–3.71), and *DNMT3A:FLT3-ITD:PTPN11* ($q = 0.02$; HR = 2.79; 95% CI: 1.41–5.5) were associated with worse outcomes while *NPM1:DNMT3A:RAD21* predicted better prognosis ($q < 0.01$; HR = 0.06; 95% CI: 0.01–0.46; Fig. 2e). Of note, the survival-associated genotypes were only slightly enriched for more frequently occurring patient genotypes (Fig. 2d), suggesting

that epistatic patterns are a weak predictor of increased leukemia aggressiveness.

Because worse survival did not associate with more frequent pairwise and triple-mutated genotypes, we wondered whether other aspects of mutation clonality might shed light on features of clinical presentation and outcomes. These observations motivated us to model the clonal landscape of AML to understand if more granular aspects of clonality (e.g., VAF or mutation ordering), rather than the simple presence or absence of mutations, might associate with features of disease.

**Variant allele frequency predicts clinical features and survival.** To investigate the relationship between mutation clonality and aspects of disease presentation and survival, we first defined high

**Fig. 2 Distinct patterns of mutation co-occurrence associate with overall survival. a** Co-occurrence and mutual exclusivity of the most frequent mutations present in de novo AML was performed using a two-sided Fisher's Exact test. Significant associations (FDR < 0.05) are colored according to the odds ratio of co-occurrence (red) or mutual exclusivity (blue). Points are sized based on the number of patients with co-occurring mutations for each genotype. **b** Summary of pairwise mutations and their association with prognosis based on Cox proportional-hazards regression modeling. Significant genotypes (FDR < 0.05) are colored according to the log-transformed hazard ratio compared to wild-type patients, with green depicting better prognosis (HR ≤ 1) and purple representing worse prognosis (HR ≥ 1). Points are sized based on the number of patients with co-occurring mutations for each genotype. **c** Scatterplot of the correlation between the odds ratio and hazard ratio of co-occurring mutations from **a**, **b**. Percentages indicate the fraction of genotypes per quadrant which associate with significant (Bonferroni FDR < 0.05) survival associations. For each error band, the measure of center is the line of best fit as derived from linear regression between the odds ratio and hazard ratio for each group. Shaded bands represent 95% confidence intervals for each linear regression. Points are sized based on the number of patients with co-occurring mutations for each genotype and colored according to the log-transformed hazard ratio compared to wild-type patients, with green depicting better prognosis (HR ≤ 1) and purple representing worse prognosis (HR ≥ 1). **d** Frequency distribution of the number of de novo patients with the most frequent 3-way mutation combinations. Bars are colored based on the association with a significant survival correlation ($p ≤ 0.05$) compared to patients with only two genes mutated: red = a significant survival association, gray = no significant association. **e** Forrest plot (left) and Kaplan–Meier plots (right; Bonferroni FDR ≤ 0.15) depicting survival analysis between triple-mutated and double-mutated genotypes. For the forest plot (left), points represent the hazard ratios calculated between triple vs. double-mutated patients using a Cox proportional-hazards model. Significant genotypes (two-sided log-rank $p ≤ 0.05$) are colored: green represents cases where the presence of all three mutations correlated with improved survival, while purple hits represent genotypes where all three mutations correlated with worse survival. q-values were calculated in terms of the false discovery rate using Bonferroni correction. Points are sized relative to the number of patients with all three mutations and bars represent the 95% confidence intervals of the hazard ratios. Source data are provided as a Source Data file.

and low VAF thresholds for each mutation based on the median copy number-corrected VAF for frequent mutations (Fig. 3a and Supplementary Fig. 4a). We observed that stratification based on median VAF for many genotypes correlated strongly with WBC counts, LDH levels, and the abundance of blasts in the peripheral blood (Fig. 3b). Interestingly, high VAF for *SF3B1*, but not other spliceosome genes, was associated with increased WBC counts ($q = 0.09$; Effect size = 0.71; 95% CI: 0.11–1.30; Fig. 3b), suggesting potential phenotypic differences for mutations in distinct spliceosome components. Interestingly, many of the mutation-phenotype associations identified using VAFs were not apparent when considering the categorical presence or absence of mutations (compare Fig. 3b and Supplementary Fig. 2). Of particular interest, we noticed pronounced differences in the directionality of effect sizes when comparing binary to VAF associations with clinical features (Fig. 3c). For example, the presence of *TP53* mutations correlated with low WBC and PB blast percentages, yet high *TP53* VAF was associated with increased WBC and PB blast percentages ($q < 0.01$; Fig. 3c). In contrast, both the presence of and higher VAF in *IDH1* and *NPM1* mutations associated with increased PB blast percentages and WBC count, respectively ($q < 0.01$; Fig. 3c). Together, these results indicate that the VAF of mutations can have additive, neutral, or re-stratifying impact on the effect size of mutational associations with clinical features of disease presentation, presumably reflecting an influence of clonal composition on disease behavior.

We next investigated the correlation of mutation VAF in risk stratification. Using a previously reported threshold of >30% to define high VAF[22], we performed survival analysis for the most frequent mutations in de novo AML patients. Univariate Cox proportional-hazards regression modeling identified four genes (*BCOR*, *KRAS*, *U2AF1*, and *NRAS*) which showed significantly worse outcomes with high VAF ($q < 0.3$; Supplementary Fig. 4b). Because a heuristic threshold applied to all genotypes might miss significant gene-specific correlations (Supplementary Fig. 4b, c), we next determined optimal thresholds for each mutation using maximally selected rank statistics (Supplementary Fig. 4d, e). Using this approach, we found that VAF thresholds could significantly re-stratify outcomes for 8 mutations ($q < 0.1$; Fig. 3d). For most mutations (*NF1*, *BCOR*, *PHF6*, *ASXL1*, *KRAS*, *PTPN11*, and *NRAS*), increased VAF associated with worse outcomes ($q = 0.01$–0.07; Fig. 3d, e and Supplementary Fig. 4e). However, for *GATA2*, higher VAF correlated with better outcomes ($q = 0.04$; HR = 0.23; 95% CI: 0.07–0.68; Fig. 3d, e and

Supplementary Fig. 4e). These analyses provide evidence that VAF adds additional information for understanding clinical features of disease presentation and risk stratification in AML. These results prompted us to investigate how clonal relationships between mutations might associate with outcomes.

**Modeling the clonal dominance of co-occurring mutations and their relationships to survival.** Experimental and computational methodologies have revealed strong trends in the ordering of mutation functional categories during leukemogenesis, with initiating mutations occurring in genes that regulate the epigenome, followed by mutations in genes involved in regulating proliferation[3,8]. Because of these trends, we hypothesized that atypical ordering of mutation acquisition might associate with differential leukemia phenotypes. Indeed, recent evidence in myeloproliferative neoplasms (MPN) and MDS supports the idea that the order of mutation acquisition can have significant effects on disease development and stratification of patient survival[45,46]. To test this hypothesis in AML, we leveraged our cohort size to infer the order of mutation acquisition for co-occurring pairs of mutations by comparing their VAF relationships (Fig. 4a and Supplementary Fig. 5). For pairwise genotypes with enough patients with nonambiguous ordering ($n = 27$), we performed survival analysis based on the putative order of mutational acquisition. We observed that the order of co-occurring *NRAS* and *GATA2* mutations robustly stratified patient outcomes; patients where *NRAS* occurred before *GATA2* showed remarkably poor survival compared to patients where *NRAS* occurred later ($q = 0.05$; HR = 0.1; 95% CI: 0.02–0.06; Fig. 4b, c). Given their pairwise VAF relationships (Fig. 4c), this result is consistent with our previous observations that high VAF in *NRAS* associates with poor outcomes ($q = 0.06$; HR = 1.56; 95% CI: 1.11–2.2) yet high *GATA2* VAF associates with improved outcomes ($q = 0.06$; HR = 0.22; 95% CI: 0.07–0.68; Fig. 3d).

Next, to increase the number of patients available for survival analysis, we grouped mutations into functional categories. Using a Bradley–Terry model, we rank-ordered mutations based on their relative order of acquisition and observed similar trends as previous reports: epigenetic dysregulation typically occurs early while mutations enhancing proliferation occur late in tumor development (Fig. 4d). We then performed survival analysis between patient groups with nonambiguous ordering of categories ($n = 12$) and identified two cases where the ordering of mutations in functional classes significantly stratified survival ($q = 0.2$; Fig. 4e and

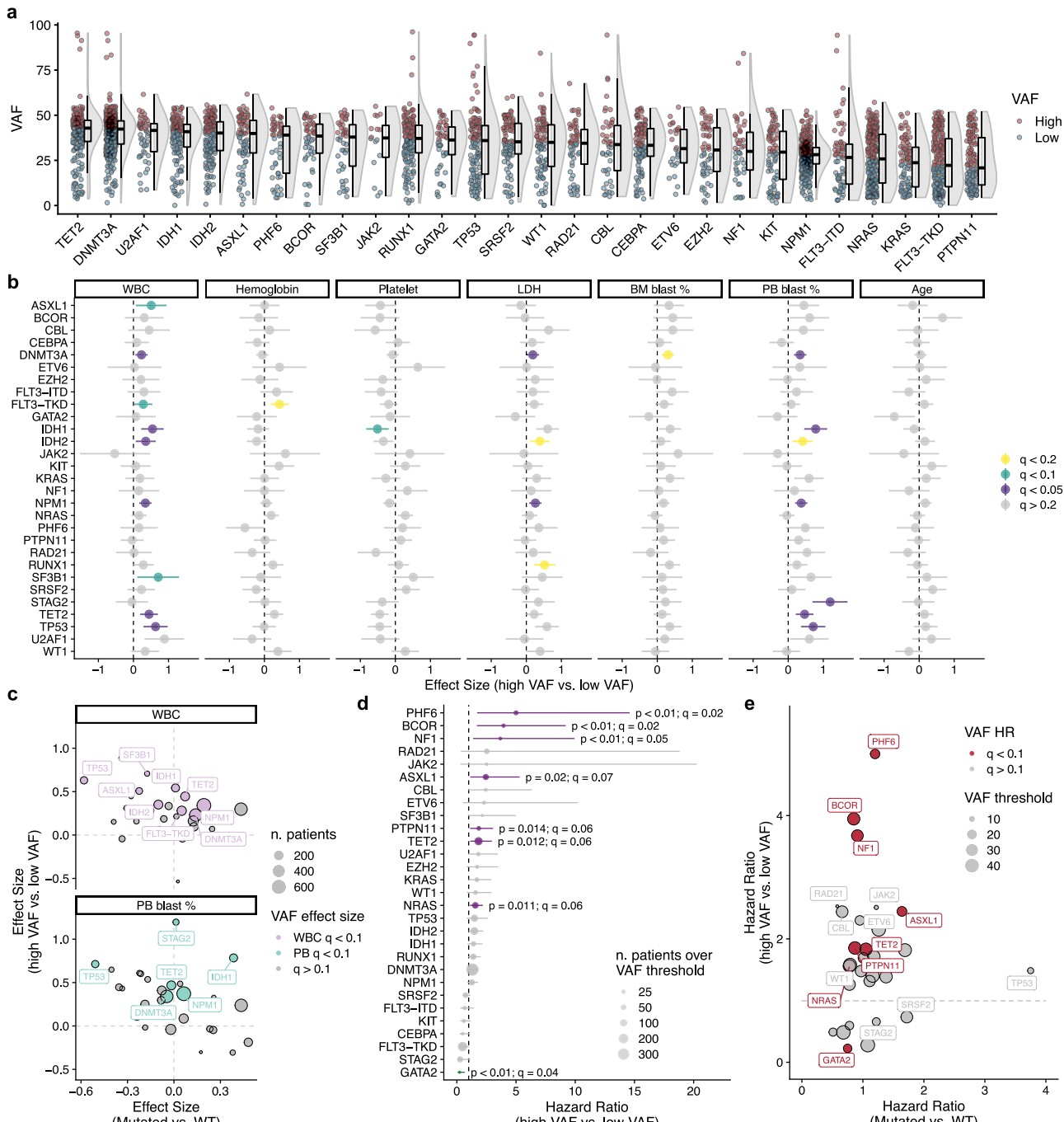

Supplementary Fig. 6). The strongest association was observed in patients with co-occurring mutations in *NPM1* and the chromatin/cohesin complex (Fig. 4e). In these patients, if a chromatin/cohesin mutation occurred before an *NPM1* variant, there was a strong association with poor survival ($q = 0.2$; HR = 1.65; 95% CI: 1.08–2.52; Fig. 4e, f). Similarly, when transcription factor mutations occurred before splicing mutations, patients showed worse outcomes ($q = 0.2$; HR = 1.43; 95% CI: 1.01–2.04; Fig. 4e, g). These results provide evidence in human data that the ordering of mutations carries prognostic significance in AML.

**Architecture of clonal evolution associates with outcomes.** Recent reports have suggested that clonal heterogeneity (i.e., mutational/clonal burden and Shannon diversity index) is

prognostic in AML[8,15,16]. To investigate similar features in our cohort, we analyzed samples profiled with whole exome sequencing (WES; $n = 731$) to model the clonal architecture of AML (Fig. 5a). Using PyClone[47], a single-cell-validated statistical inference method to model clonal population structure, we identified distinct clonal genotypes and cancer cell fractions for each sample across our cohort (Fig. 5b and Supplementary Fig. 7a, b). We confirmed that higher mutation burden correlates with worse survival ($p = 0.015$; Supplementary Fig. 7c); however, in contrast to a previous report[29], we found no association between clonal burden and outcomes ($p = 0.51$; Supplementary Fig. 7d). We also observed no statistical correlation between mutational or clonal burden with age or European LeukemiaNet (ELN) risk (Kruskal–Wallis $p = 0.14$; Supplementary Fig. 7e–h). Not surprisingly, a robust correlation between mutation burden

**Fig. 3 Variant allele frequency associates with features of disease presentation and overall survival. a** VAF distribution for frequent mutations in our de novo cohort (*n* = 1636 patients). Mutations were assigned as high or low VAF based on the median copy number-corrected VAF for each gene. Red points represent cases where the VAF is above the median while blue points represent those below the median VAF. For each distribution, the boxplot represents the boundaries for the first and third quartiles with a line at each median; whiskers delimit the highest data point below the third quartile +1.5× the interquartile distance and the lowest data point above the first quartile −1.5× the interquartile distance. **b** Distribution of effect sizes for differences in clinical features of AML based on high or low VAF for each mutation. Points represent the effect size (Cohen's *d*) between high and low VAF for each genotype across the different clinical variables (*n* patients: WBC = 1507; Hemoglobin = 1280; Platelet = 1292; LDH = 1228; BM blasts = 1472; PB blasts = 1385; Age = 1643). Significant associations are colored based on the level of significance (Bonferroni FDR < 0.2); error bars represent the 95% confidence intervals of the effect sizes. **c** Scatterplots of effect sizes for WBC levels and peripheral blood blast percentages between mutated and wild-type patients versus effect sizes calculated between high and low VAF for each mutation. Points are sized based on the number of patients analyzed and colored based on VAF effect size significance (FDR < 0.1). **d** Forest plot summarizing univariate Cox proportional-hazards regression modeling of common mutations based on VAF thresholds in the de novo cohort. Points represent the hazard ratio for overall survival between high and low VAF groups based on VAF thresholds calculated using maximally selected rank statistics. Points are sized based on the number of patients above the VAF threshold. Error bars represent the 95% confidence intervals of the hazard ratios. Green hits represent cases where higher VAF correlated with improved survival, while purple hits represent genotypes where increased VAF correlated with worse survival. **e** Scatterplot of hazard ratios calculated between mutated and wild-type patients versus hazard ratios calculated between high and low VAF for each mutation. Hazard ratios are calculated using a standard Cox proportional-hazards model. Points above the dotted line indicate mutations where greater VAF associates with worse outcomes compared to patients with lower VAF for that mutation. Points are colored by significance of VAF hazard ratio calculations (red points = Bonferroni FDR < 0.1) and sized relative to the VAF threshold for each genotype. Source data are provided as a Source Data file.

and the number of unique clones was observed (Kruskal–Wallis *p* < 0.001; Fig. 5c). Because there was a strong correlation between mutation and clonal burden, yet only mutation load associated with survival (Supplementary Fig. 7c, d), we sought to understand if the distribution of mutations across clones associated with outcomes. Indeed, higher median mutation burden per clone was correlated with poor prognosis (*p* = 0.013; Fig. 5d), suggesting that the accumulation of mutations in the same clonal population associates with increased leukemia fitness.

To further characterize the architecture of these leukemias, we modeled clonal evolution trajectories using ClonEvol (Fig. 5a and Supplementary Fig. 7i)[48]. We observed that most tumors displayed linear trajectories rather than branched evolution (Supplementary Fig. 7j), in agreement with a recent report[29]. AML patients exhibiting branched evolution showed increased mutational and clonal burden compared to tumors with linear evolution (*p* < 0.001; Supplementary Fig. 7k, l). Strikingly, patients exhibiting branched clonal evolutionary architectures showed significantly better overall survival, despite having an increased mutational burden (*p* = 0.029; Fig. 5e, f and Supplementary Fig. 7k). Interestingly, clonal diversity, as measured by multiple metrics, did not associate with outcomes (Shannon diversity index *p* = 0.31; MATH score *p* = 0.17; Supplementary Fig. 7m, n). However, we observed that high mutational burden re-stratified outcomes in patients exhibiting branched evolution (*p* = 0.007) but that this association was not seen in patients with linear evolution (Fig. 5g and Supplementary Fig. 7o, p). Specifically, low mutational (but not clonal) burden in branched tumors identified a low-risk patient subset (*p* < 0.001; Fig. 5g and Supplementary Fig. 7o, p). These results reveal a unique interplay between clonal heterogeneity and leukemia fitness (Fig. 5h) and define a low-risk subset of patients.

**Subclonal abundance predicts drug sensitivity in patient samples.** Previous work modeling mutation-specific drug sensitivity has used the presence or absence of mutations, not their clonal abundance, to correlate genomic alterations with drug response[9]. To investigate if the clonality of mutations correlate with drug sensitivity, we analyzed data from an ex vivo drug screen of primary AML samples[9] and modeled the correlation between VAF and drug response (measured using area-under-the-curve; AUC) in de novo samples (Fig. 6a). Differential drug sensitivity analysis between wild-type and mutated samples identified expected correlations between *FLT3* status and tyrosine

kinase inhibitor (TKI) response in addition to *NRAS*-dependent resistance, amongst other associations (Fig. 6b). Analyzing the copy number-corrected VAF landscape of recurrent mutations showed significant variability in clonal abundance (Fig. 6c and Supplementary Fig. 8a), suggesting a potential range of drug sensitivity based on the subclonal prevalence of specific mutations. For drug-gene pairs with sufficient heterogeneity of VAF and drug response (see Methods), linear regression of drug sensitivity against VAF identified multiple cases where VAF correlated strongly with sensitivity to targeted agents (Fig. 6d and Supplementary Fig. 8b–d). We noted unique trends for the most predictive genes; higher VAF of *IDH1* and *NPM1* showed only increased sensitivity to drugs, while greater *NRAS* VAF showed only increased resistance to multiple agents (Fig. 6d). One of the strongest correlations between VAF and drug sensitivity was between ponatinib and *IDH1*, where a VAF increase of 35% correlated with a drop in AUC of 127 ($R^2$ = 0.69; *p* = 0.003; Fig. 6e). Conversely, one of the strongest correlations between VAF and drug resistance was between pelitinib and *NRAS*, where a VAF increase of 42% correlated with an increase in AUC of 176 ($R^2$ = 0.59; *p* = 0.001; Fig. 6f). Of particular interest, across the panel of FLT3-specific or general TKIs, *FLT3-TKD* VAF showed significantly more AUC-VAF correlations than *FLT3-ITD* (Fig. 6d and Supplementary Fig. 8d). Because patient survival based on *FLT3* status is highly dependent on the presence of additional mutations[8], we re-analyzed *FLT3* VAF-dependent sensitivity in the context of common mutational backgrounds. Co-occurrence of *FLT3-ITD* with *DNMT3A* predicted resistance to axinatinib, cediranib, crizotinib, ponatinib, and tofacitinib in an ITD VAF-dependent manner (Fig. 6g). These results indicate that the subclonal prevalence of secondary mutations in AML might predict response to targeted therapy.

When we compared the drug-gene correlations identified through our binary and VAF analyses in the de novo cohort, we saw no overlap in drug-gene pairs (Supplementary Fig. 8e), indicating that modeling drug sensitivity using VAFs identified unique associations between AML mutations and targeted agents. Of particular clinical significance, we observed that *NRAS* VAF was a strong predictor of drug resistance in both de novo and secondary AML samples (Supplementary Figs. 8d and 9), despite the fact that secondary samples are inherently far more resistant to the same set of targeted agents compared to de novo samples[9]. Given that *NRAS* is the fourth most common mutation in AML, these results suggest a previously unappreciated biomarker for

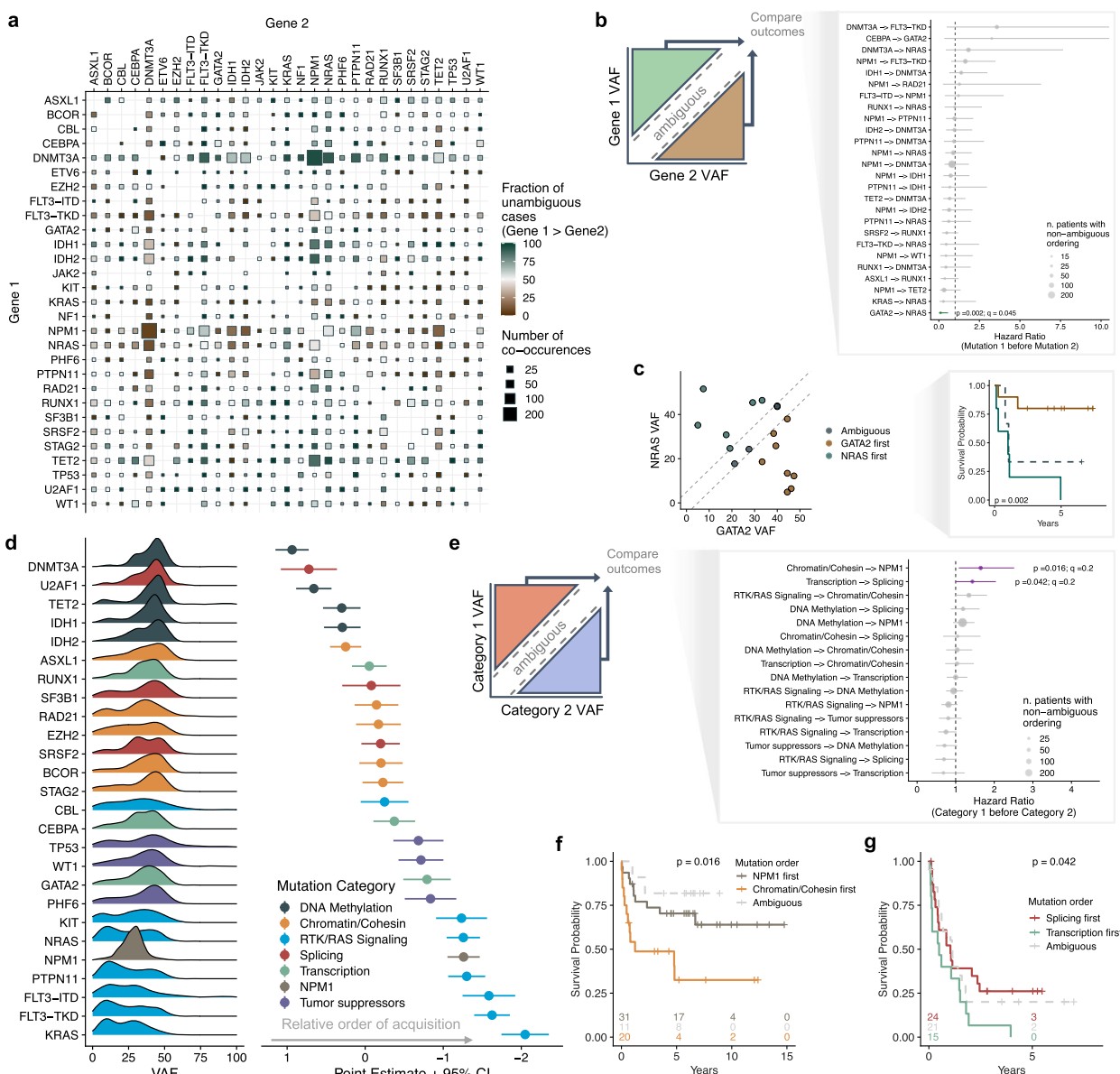

**Fig. 4 Clonal dominance of co-occurring mutations stratifies survival. a** Summary of pairwise clonal relationships for recurrent mutations present in de novo AML. Color represents the fraction of patients where mutations in gene 1 occurred before gene 2 (i.e. are clonally dominant; green = gene 1 before gene 2; brown = gene 1 after gene 2). Size is scaled to reflect the total number of patients with co-occurring mutations. **b** Schematic depicting how pairwise VAF relationships were used to bin patients into groups (left) and forest plot of hazard ratios calculated from univariate Cox proportional-hazards modeling based on the order of pairwise mutations (right). Points represent the hazard ratio, as calculated using standard Cox proportional-hazards regression, between patients with different mutation order. Bars represent the 95% confidence intervals of the hazard ratios and points are sized based on the number of patients with defined mutation ordering. **c** Scatterplot (left) and Kaplan–Meier plot (right) showing how the order of mutation acquisition in patients with co-occurring mutations in *NRAS* and *GATA2* robustly improved patient stratification. Green points/line represent cases where *NRAS* mutations occur before those in *GATA2*. Brown points/line represent cases where *GATA2* mutations occur before those in *NRAS*. The reported *p*-value was calculated using a two-sided log-rank test. **d** A Bradley–Terry model was used to assign the relative order of global mutation acquisition from pairwise relationships as determined in **a**. Only patients with at least two mutations were considered in this model. Density plots (left) represent the VAF distribution for each mutation in the analysis (corrected for copy number and X-linkage in males) and are ordered on the *y*-axis based on their relative order of acquisition compared to all other genes in the analysis. Points and error bars (right) represent the Bradley–Terry model results for the point estimate and 95% confidence interval, respectively, for relative gene ordering in temporal acquisition. **e** Schematic depicting how pairwise VAF relationships were used to bin patients into groups (left) and forest plot of hazard ratios calculated from univariate Cox proportional-hazards regression modeling based on the order of mutation category acquisition (right). Bars represent the 95% confidence intervals of the hazard ratios. **f**, **g** Kaplan–Meier plots for significant pairs from **e**. *p*-values in **c**, **f**, **g** were calculated between nonambiguous groups using a two-sided log-rank test. Source data are provided as a Source Data file.

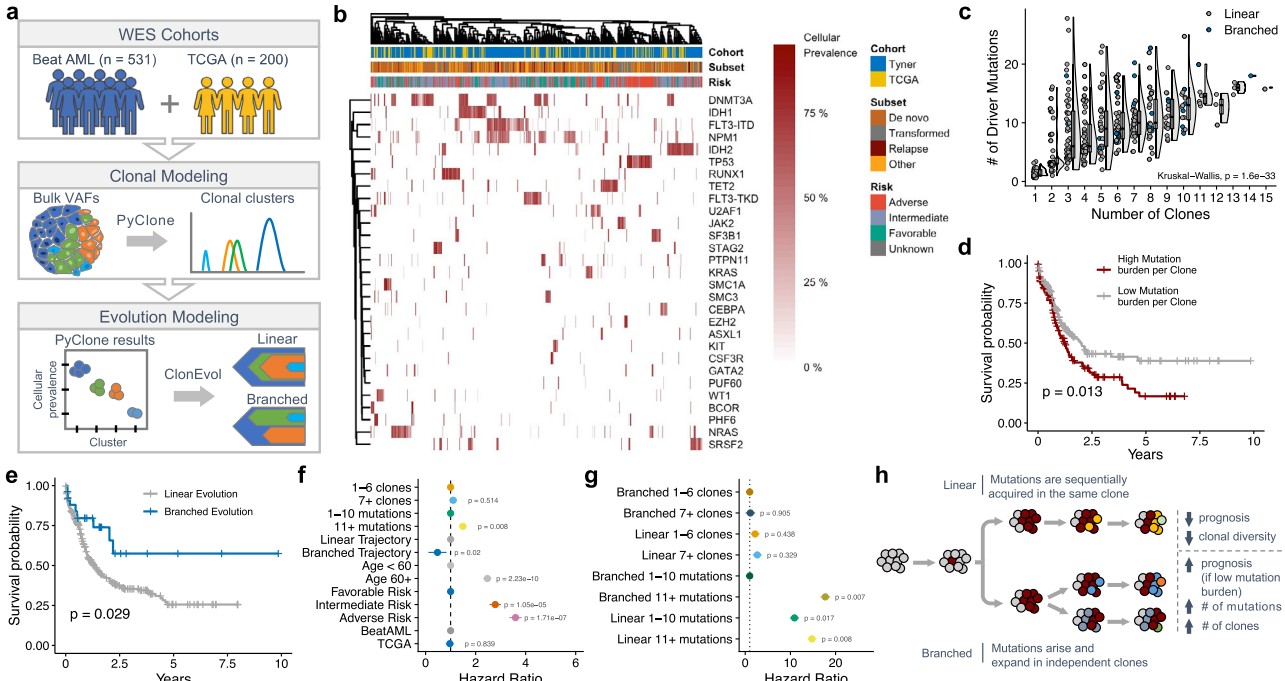

**Fig. 5 Architecture of clonal evolution associates with survival outcomes. a** Schematized workflow for modeling clonal architecture in a cohort of WES patients. Briefly, a deep-sequenced cohort was assembled and analyzed using PyClone to generate robust clonal populations and cellular prevalences. These cellular prevalence estimates were then leveraged to model the temporal acquisition of mutations and clonal architecture using ClonEvol. **b** Heatmap summarizing PyClone results of the per-patient cellular prevalence for the most common clonal genotypes. Each column is an individual patient sample grouped by hierarchical clustering based on similarity in clonal patterns. For patients with multiple mutations in the same gene, only the mutation with the largest cancer cell fraction (CCF) is shown. **c** Correlation between mutation burden and the number of unique clones derived from PyClone in the de novo cohort (Kruskal–Wallis $p = 1.6e{-}33$; $n = 409$ patients). Points are colored by broad clonal evolution architecture as determined by ClonEvol (blue = branched evolution; gray = linear evolution). For each distribution, the boxplot represents the boundaries for the first and third quartiles with a line at each median; whiskers delimit the highest data point below the third quartile $+1.5\times$ the interquartile distance and the lowest data point above the first quartile $-1.5\times$ the interquartile distance. **d** Kaplan–Meier plot showing the association between higher median mutational burden per clone (red curve) and worse outcomes in de novo patients (two-sided log-rank test). **e** Kaplan–Meier plot showing the association of improved outcomes in patients exhibiting branched evolutionary architecture (blue curve = branched evolution; gray curve = linear evolution; two-sided log-rank test). **f** Forrest plot depicting univariate Cox proportional-hazards ratios for various aspects of the clonal architecture analyses. **g** Forrest plot depicting univariate Cox proportional-hazards ratios for clonal and mutational burden risk stratification based on linear or branched architecture. **h** Schematic depicting the different genetic and clinical features associated with evolutionary architecture. Source data are provided as a Source Data file.

resistance to current and emerging targeted therapies. In aggregate, these results identify multiple drug-gene sensitivity relationships which warrant further experimental validation and retrospective analyses of therapy response in clinical trials for AML.

## Discussion

Here, we report a large, aggregated cohort of AML profiled by deep sequencing and describe several unique features for risk stratification and prediction of sensitivity to a panel of small molecule inhibitors. Of particular interest, we observed unreported associations between VAF and the architecture of clonal evolution in leukemia with drug response and clinical outcomes. These observations suggest an unappreciated nuance in how genotypically similar patients might differ in disease risk and clinical response.

Our results suggest that VAF is a clinically useful feature for improving risk stratification in specific AML genotypes. This observation may reflect differences in the timepoint of therapeutic intervention rather than underlying biology of the disease. For example, our observation that patients with low *NRAS* VAF have better prognosis than those with high *NRAS* VAF may reflect that low VAF patients were diagnosed and treated earlier in disease progression. However, our observation that *NRAS* VAF was a

strong predictor of drug resistance offers support for the hypothesis that the biological properties of specific subclonal genotypes might drive VAF risk stratification. Nevertheless, the underlying biological reason as to why the VAF of only certain mutations enhances risk stratification remains to be understood.

Our observation that the ordering of specific mutations and functional categories carries prognostic significance might reflect underlying differences in leukemia aggressiveness given the type of initial mutation. Indeed, recent functional and observational studies in MPN, MDS, and mesothelioma have provided some of the earliest evidence for differences in cell fitness and tumorigenic potential given the ordering of mutations and clonal architecture[45,46,49]. Because there are strong trends in the ordering of functional categories of mutations in AML, it is feasible that this order is preferentially selected for in tumor evolution because it provides a fitness advantage in disease progression. However, even with our cohort size, analyzing survival by mutation order remains weakly powered for less frequent genotypes and survival differences based on the ordering of mutation categories represent relatively few patients. Additionally, using VAF to infer mutation co-occurrence and clonality may not always be accurate, and more refined single cell genotyping could be employed to more robustly define mutational ordering and correlate clonal diversity with outcomes[28–30].

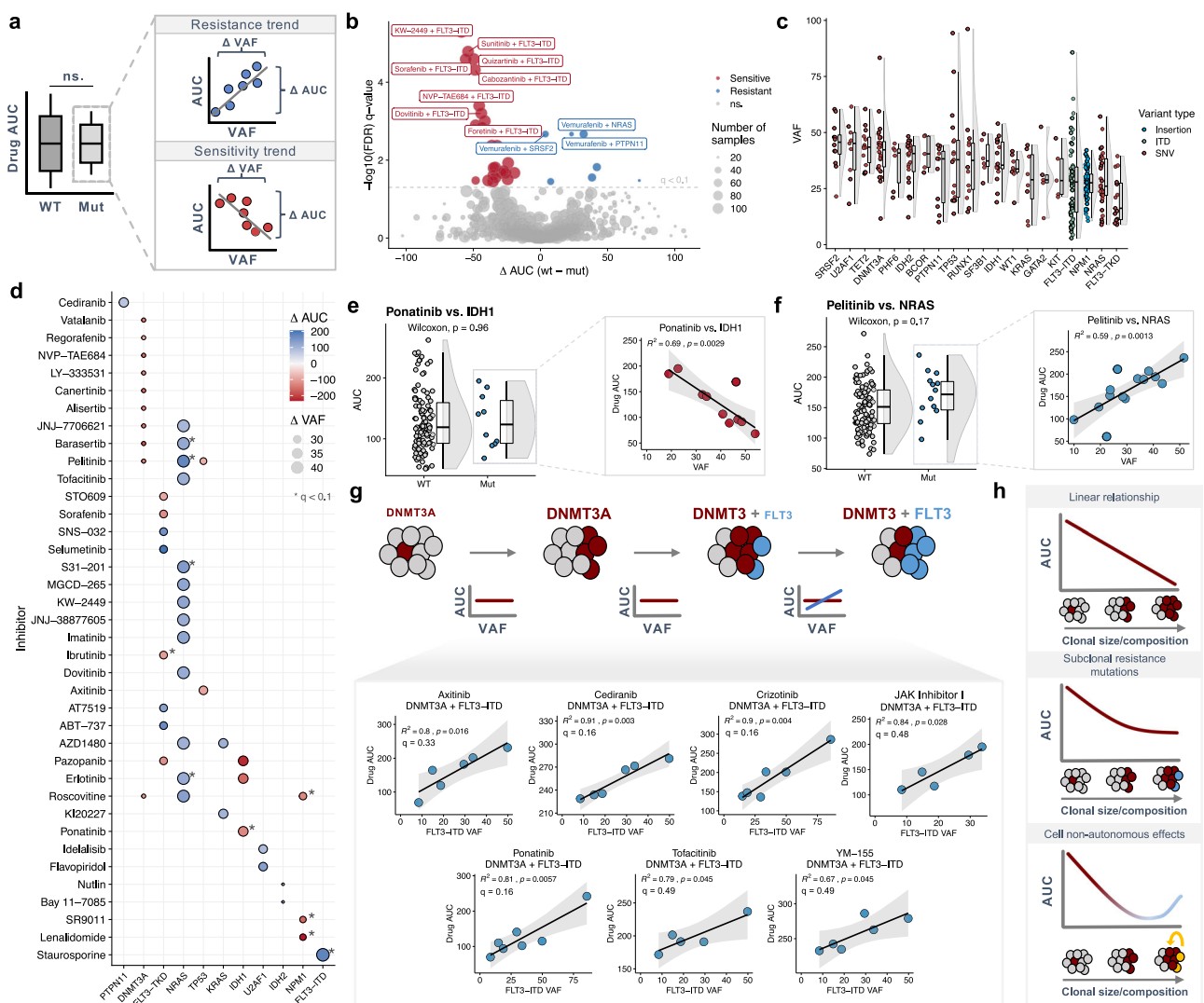

**Fig. 6 Clonal abundance predicts drug sensitivity in primary AML samples. a** Schematic depicting how linear regression of drug response (AUC) against VAF can identify correlations between drug sensitivity and the clonal prevalence of mutations. Red points represent sensitivity trends while blue points represent resistance trends. **b** Volcano plot of drug response between mutated and wild-type samples for de novo samples from the Beat AML study. Points are sized based on the number of samples analyzed and colored by significance (Bonferroni FDR < 0.1; red = sensitive, blue = resistant). **c** Copy number-corrected VAF distribution for mutations with paired drug data in the de novo cohort of the Beat AML study ($n^{mut + drug} \geq 5$; $n = 202$ biologically independent patient samples). For each distribution, the boxplot represents the boundaries for the first and third quartiles with a line at each median; whiskers delimit the highest data point below the third quartile +1.5× the interquartile distance and the lowest data point above the first quartile −1.5× the interquartile distance. **d** Dotplot of the most significant ($p < 0.05$) drug-gene correlations identified through linear regression of drug AUC against mutation VAF in de novo AML samples. Points are sized based on the range of VAFs for each mutation and are colored based on the type of drug sensitivity trend (red—sensitive; blue—resistant). Asterisks represent drug-gene associations with a Bonferroni FDR < 0.1. **e, f** Representative binary distributions (left) and AUC-VAF scatterplots (right) for clinically relevant sensitivity and resistance VAF correlations for IDH1 (**e**; $n^{WT} = 195$ samples; $n^{Mut} = 10$ samples) and NRAS (**f**; $n^{WT} = 177$ samples; $n^{Mut} = 14$ samples), respectively. For each distribution (left), the boxplot represents the boundaries for the first and third quartiles with a line at each median; whiskers delimit the highest data point below the third quartile +1.5× the interquartile distance and the lowest data point above the first quartile −1.5× the interquartile distance; p-values are calculated using a two-sided Wilcoxon rank-sum test. For each scatterplot (right), shaded bands represent 95% confidence intervals for each linear regression. For each error band, the measure of center is the line of best fit as derived from linear regression between the drug AUC and VAF for each mutation-drug pair. **g** Schematic (top) depicting the potential correlation between the subclonal prevalence of a secondary mutation (e.g. FLT3-ITD) and sensitivity to inhibitors. AUC-VAF scatterplots (bottom) for pairwise genotypes with enough samples ($n \geq 5$; DNMT3A:FLT3) where linear regression of drug AUC against VAF revealed strong resistance trends. Shaded bands represent 95% confidence intervals for each linear regression. For each error band, the measure of center is the line of best fit as derived from linear regression between the drug AUC and VAF for each mutation-drug pair. **h** Schematics representing possible relationships between VAF and drug response. Source data are provided as a Source Data file.

Future studies in even larger cohorts will be needed to address this question more precisely.

Understanding the interplay between tumor heterogeneity and patient risk can improve our understanding of cancer biology and clinical intervention. By modeling the trajectories of clonal evolution, we show that the serial accumulation of mutations in the same clone (linear evolution), rather than their distribution in multiple subclones (branched evolution), correlates with poorer prognosis. This observation could be explained by the mechanistic hypothesis that sequential acquisition of mutations in the

same clone progressively increases cell-autonomous leukemia robustness leading to therapeutic resistance. Additionally, the effects of mutation co-occurrence could be due to either cell-autonomous or cell non-autonomous effects, even for mutations that occur in separate clones. Overall, these findings indicate a previously unappreciated nuance of what type of clonal heterogeneity more accurately associates with clinical outcomes.

Biomarkers of drug sensitivity are of significant clinical relevance. Here, we present evidence that the VAF of specific mutations in AML is predictive of drug sensitivity to a wide array of small molecule inhibitors in various stages of clinical development. Critically, the drug-gene correlations identified in our approach showed no overlap with relationships identified in a previous analysis of the same dataset[9]. This is most likely because we (i) analyzed de novo and secondary samples independently and (ii) utilize VAF as a predictive feature in this drug screening dataset. One caveat of our approach is the assumption of steady clonal/subclonal structures during ex vivo maintenance. The question of why some mutations show VAF correlation to drug sensitivity and others do not is fascinating and warrants careful experimental study. Possible contributing factors could be the presence of additional co-occurring resistance mutations or non-autonomous cellular interactions (Fig. 6h). Importantly, many VAF-drug response relationships are not evident with a straightforward binary comparison (e.g., Fig. 6e, f) and would be missed by traditional analyses. Further experimental follow-up and retrospective analyses of drug response and VAF in clinical trials are warranted to identify biomarkers of response to current and emerging therapies

Retrospective analyses in large, aggregated cohorts are not able to define causal relationships between molecular or clinical observations and the underlying mechanistic biology of cancer. Regardless, they provide comprehensive descriptions of disease features and can identify associations not observed in smaller studies. Although raw sequencing data was not available to uniformly analyze all samples in our study, intra-patient VAF relationships should be maintained regardless of the inter-patient sequencing depth or platform, thus mitigating traditional issues with aggregating preprocessed data. Additional studies are required to validate our findings, as well as orthogonal data sets such as single-cell proteomics and transcriptomics. In particular, experimental models of co-occurring mutations and controlled mutation timing will be important to understand how these features drive leukemogenesis and evolution. As larger cohorts of clinically annotated deeply sequenced AML emerge, coupled with large single-cell studies[28–30,50], validation of our molecular correlations to clinical outcomes will become feasible. Retrospective analysis of mutation VAF and response to targeted therapies in clinical trial cohorts will be essential to validate our observations of VAF-dependent drug responses.

## Methods

**Study design, cohort aggregation, and data homogenization.** We performed a systematic literature review for cohorts of sequenced adult AML patients ($n \geq 50$), excluding studies focused on specific genotypes or AML subtypes (e.g., secondary AML, normal karyotype, single genotype, etc.) in order to maintain a broadly representative landscape of the disease. For most studies, sequencing results and clinical annotations were downloaded from published supplemental data tables. In one case, additional data (e.g. VAFs, sex) were obtained through direct communication with the authors[37]. Molecular and clinical annotations were manually assessed and modified when necessary to generate uniform coding across studies. When available, raw VAFs were corrected using copy number status or karyotype data. In total, we aggregated 13 studies into our cohort (Supplementary Fig. 1a).

**Clonal modeling and diversity analysis.** Rigorous clonal modeling was performed using PyClone (v0.13.0; Python v2.7)[47] for all samples with whole exome sequencing results ($n^{total} = 731$; $n^{de\ novo} = 329$); all variants (rather than just driver

mutations) were used to enhance the modeling. For samples analyzed by multiple mutation callers, only consensus calls were used as inputs to PyClone, however, reads mapped by Pindel were included because it is exclusively used to call *FLT3-ITD* indels. Raw VAFs were corrected using available copy number and karyotype information prior to clonal modeling (Supplementary Fig. 5a). Clonal architecture was modeled using the R package ClonEvol (clonevol_0.99.11) to define types of clonal evolution (branched or linear)[48]. Shannon diversity index for each tumor was calculated using the cellular prevalence output obtained from PyClone or VAF (R package: vegan_2.5-7).

**Mutation order analysis.** We defined clonal dominance/mutational ordering based on an empirical VAF threshold. Iterating through a VAF threshold from 1-10% showed minimal differences in pairwise mutation ordering or Bradley–Terry modeling. Therefore, to reduce the number of ambiguous cases and also retain as many high confidence calls as possible, a mutation was defined as occurring earlier (clonally dominant) than a second mutation if its VAF was $\geq 5\%$ higher. Mutations within 5% VAF were assigned as ambiguous in their relative ordering. Using this criterion for pairwise mutation ordering, we applied a Bradley–Terry model (R package: BradleyTerry2_1.1-2) to all de novo samples with at least two mutations to generate point estimates and 95% confidence intervals for relative order of acquisition. Orderings of pairwise mutation categories (e.g., tumor suppressors, transcription factors) were defined using a similar approach. For cases with multiple mutations in the same gene/category, only the largest VAF per gene/category was used to assign ordering.

**Drug sensitivity analysis.** Drug sensitivity and whole exome sequencing results were available for 122 small molecule inhibitors across 168 de novo patients from the Beat AML cohort[9]. For each drug-gene combination, linear regression of drug AUC against mutation VAF was performed only if the following criteria were met: number of samples with mutation and drug data $\geq 5$; $\Delta$VAF (VAF$^{max}$ − VAF$^{min}$) $\geq 0.25$; and $\Delta$AUC (AUCmax − AUC$^{min}$) $\geq 75$ (R package: ggpubr_0.4.0). These heuristic cut-offs were implemented in order to restrict our analysis to drug-gene pairs with (i) sufficient numbers of samples and (ii) a dynamic range of both drug response ($\Delta$AUC) and clonal heterogeneity ($\Delta$VAF). Drug sensitivity was defined as interactions in which increased VAF was associated with decreased AUC; resistance was defined as interactions in which increased VAF was associated with increased AUC. To account for multiple hypothesis testing for drug-gene pairs meeting these criteria, we calculated $q$ values in terms of the false discovery rate using Bonferroni correction (R package: stats_4.0.4).

**Statistical analysis.** Overall survival was modeled with standard Cox proportional-hazards regression methods using binary mutation calls, VAF thresholds, clonality, and mutation ordering as random effects (R package: survminer_0.4.9). $P$ values from survival analyses were calculated using a log-rank test. Optimal VAF thresholds for survival analysis were determined using maximally selected rank statistics (R package: MaxStat v. 0.7-25). Odds ratio calculations and Fisher's Exact tests were used to analyze pairwise categorical variables (R package: stats_4.0.4). Effect sizes (Cohen's $d$) for differences in clinical features were calculated between mutation genotypes and high vs. low VAF groups per genotype (R package effsize_0.8.1). Multiple hypothesis testing correction was performed using Bonferroni FDR correction to calculate reported $q$ values (R package: stats_4.0.4). All tests are two-sided unless otherwise indicated. All analyses were performed using the RStudio statistical software platform v1.2.1114 and in R version 4.0.4 (2021-02-15).

**Reporting summary.** Further information on research design is available in the Nature Research Reporting Summary linked to this article.

## Data availability

Processes mutation calls and clinical data for the publicly available datasets used in this study were obtained from the following links: TCGA (http://download.cbioportal.org/laml_tcga_pub.tar.gz); Tyner (https://static-content.springer.com/esm/art%3A10.1038%2Fs41586-018-0623-z/MediaObjects/41586_2018_623_MOESM3_ESM.xlsx); Papaemmanuil (https://github.com/gerstung-lab/AML-multistage/blob/master/data/AMLSG_Clinical_Anon.RData?raw=true; https://raw.githubusercontent.com/gerstung-lab/AML-multistage/master/data/AMLSG_Genetic.txt); Lindsley (https://ashpublications.org/blood/article/125/9/1367/34220/Acute-myeloid-leukemia-ontogeny-is-defined-by); Wang (https://www.oncotarget.com/article/7028/); Au (https://diagnosticpathology.biomedcentral.com/articles/10.1186/s13000-016-0456-8); Welch (https://www.nejm.org/doi/full/10.1056/NEJMoa1605949); Garg (https://ashpublications.org/blood/article/126/22/2491/34632/Profiling-of-somatic-mutations-in-acute-myeloid); Greif (https://clincancerres.aacrjournals.org/content/early/2018/01/12/1078-0432.CCR-17-2344.figures-only?versioned=true); Hirsch (https://static-content.springer.com/esm/art%3A10.1038%2Fncomms12475/MediaObjects/41467_2016_BFncomms12475_MOESM1147_ESM.pdf; https://static-content.springer.com/esm/art%3A10.1038%2Fncomms12475/MediaObjects/41467_2016_BFncomms12475_MOESM1148_ESM.xlsx); Huet (https://ash.silverchair-cdn.com/ash/content_public/journal/blood/132/8/10.1182_blood-2018-03-840348/4/

blood840348-sup-tables2.xlsx?Expires=1579153943&Signature=VCqH4IQOtZ~
x9YPe1dJSz1AHegirMKv6N4euRcbvd3gsayA~RULg17B6mjZY0lrUzbbbj5D7MdLJN-
zI0sUk4QFDz6IKvvWsV~l1CQL4J1lOHejvckLQzXikTy-
4SD62jwmDMgSeBMhXqVo6pmdHXI0iO7b8ARJV8EDZ9Bt8h~h6nIXOaf5JZqe8he-
hacxWIMjCWRd5OulQjpqQu10MLfK6GrpISgRGDYe1WODtkPIQ~o6qQ-
j~eqTFGvzwbSmhWuRGSAFKEFzPzgYLiKmkQ8errjq9vI2uHgpurlwtB98gQOnizt9-
goXVoDPsGd0P~fk8CPPEGJdUtSkOB41b8p3JTg__&Key-Pair-
Id=APKAIE5G5CRDK6RD3PGA); Majeti (https://static-content.springer.com/esm/art
%3A10.1038%2Fng.3646/MediaObjects/41588_2016_BFng3646_MOESM43_ESM.xlsx;
https://cancerdiscovery.aacrjournals.org/highwire/filestream/41655/
field_highwire_adjunct_files/6/181878_2_supp_4135424_zsdnjx.xlsx). Raw TCGA
sequencing data used in this study are available at the NIH Genomic Data Commons
[https://gdc.cancer.gov]. Raw sequencing data for the Papaemmanuil study is deposited
at the European Genome-Phenome Archive with accession number EGAS00001000275.
For the Tyner study, sequencing data and clinical annotations can be found at dbGaP and
the Genomic Data Commons. The dbGaP study ID is 30641 and accession ID is
phs001657.v1.p1. For the Welch dataset, exome sequencing data are deposited in the
database of Genotypes and Phenotypes under the accession number phs000159. For the
Huet study, raw exome and targeted sequencing data are deposited at the European
Genome-phenome Archive (EGA, https://ega-archive.org) under accession number
EGAS00001001779 - data are available upon request from the Data Access Committee of
the MyPAC clinical research group. The aggregated mutational and clinical data frames
used for the analyses in this paper are available at https://github.com/brooksbenard/
Meta_AML. Source data are provided with this paper.

## Code availability

All code used to aggregate, analyze, and visualize data for this study has been deposited at
https://github.com/brooksbenard/Meta_AML. The Zenodo DOI is: https://doi.org/
10.5281/zenodo.5641315[51]. Source data are provided with this paper.

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

## Acknowledgements

This work was supported by the National Institutes of Health, National Cancer Institute (1R01CA251331, R.M.), the Ludwig Institute for Cancer Stem Cell Research and Medicine (R.M.), Blavatnik Family Foundation (B.A.B), and NIH training grant 5T32CA9302-40 (B.A.B. and L.B.L). R.M. is a Leukemia & Lymphoma Society Scholar. D.T. is supported by a Commonwealth Serum Laboratories Centenary Fellowship, Australian Medical Research Future Fund, National Health and Medical Research Council Ideal Grant APP1182564, and the Leukemia & Lymphoma Society/Snowdome Translational Research Program. The authors would like to thank Amy Fan and Asiri Ediriwickrema for their critical input and editing of the manuscript.

## Author contributions

B.A.B, R.M., D.T. and A.G. designed the study and wrote the manuscript. B.A.B and L.B.L performed the analysis and made the figures. B.A.B, L.B.L. and A.A. generated the data.

## Competing interests

R.M. is on the Board of Directors of BeyondSpring Inc., and Scientific Advisory Boards of Coherus BioSciences, Kodikaz Therapeutic Solutions Inc., and Zenshine Pharmaceuticals. R.M. is an equity holder and founder of CircBio Inc. and Pheast Therapeutics Inc.. R.M. is an inventor on a number of patents related to CD47 cancer immunotherapy licensed to Gilead Sciences, Inc. The numbers and titles of the awarded patents related to CD47 are as follows: U.S. Patent No. 8,562,997 *"Methods of Treating Acute Myeloid Leukemia by Blocking CD47"*; U.S. Patent No. 8,709,7429; 9,193,955; 9,796,781; 10,662,242 *"Markers of Acute Myeloid Leukemia Stem Cells"*; U.S. Patent No. 8,758,750 *"Synergistic Anti-CD47 Therapy for Hematologic Cancers"*; U.S. Patent No. 9,017,675; 9,382,320 *"Humanized and Chimeric Monoclonal Antibodies to CD47"*; U.S. Patent No. 9,399,682; 9,493,575; 9,605,0769,611,329; 9,624,305; 9,765,143; 10,640,561 *"Methods of Manipulating Phagocytosis Mediated by CD47"*; U.S. Patent No. 9,623,079 *"Methods for Achieving Therapeutically Effective Doses of Anti-CD47 Agents for Treating Cancer"*; U.S. Patent No. 10,087,257; 10,487,150 *"SIRP Alpha-Antibody Fusion Proteins"*; U.S. Patent No. 10,301,387 *"Methods for Achieving Therapeutically Effective Doses of Anti-CD47 Agents"*. The numbers and titles of current patent applications related to CD47 are as follows: US-2017210803 *"Treatment of Cancer with Combinations of Immunoregulatory Agents"*; US-2020048369 *"Modified Immunoglobulin Hinge Regions to Reduce Hemagglutination"*; US-2020147212 *"Dosing Parameters for CD47 Targeted Therapies in Hematologic Malignancies"*; PCT/US2021/024937 *"Pharmaceutical Formulation of Hu5F9-G4 for Human Therapeutic Use"*. The remaining authors have no competing interests to disclose.
