## [Peer review file · Nature Communications]

REVIEWER COMMENTS

Reviewer #1 (Remarks to the Author): Expert in AML genomics, evolution and therapy

Manuscript: Clonal architecture is associated with clinical outcomes and drug response in acute myeloid leukemia

Following the advent of next generation sequencing technologies, bulk and single-cell DNA sequencing of human AMLs has revealed that clonal heterogeneity is a prevalent feature of disease. Previous work has pointed to mutational burden and clonal diversity as informative measures that predict clinical outcomes. Here, Benard et al. perform a meta-analysis of 12 AML bulk-sequencing cohorts with increased granularity to determine whether variant allele frequency, mutation co-occurrence, timing of mutation acquisition, and clonal evolutionary patterns are predictive of phenotypic changes, drug sensitivity, and survival outcomes. The authors clearly prioritize statistical power in this comprehensive analysis of sequencing data and deliver interesting results that require functional validation moving forward. However, there are several questions that should be addressed and suggested analyses that the reviewer believes would strengthen the study.

Questions and Comments

1. Fig 1 and Supp. Fig 1— Since this meta-analysis includes many different cohorts from different institutions, one concern that arises is that treatment history is probably heterogeneous. This is likely to influence leukemia fitness and may lead to subclonal selection in some specific contexts. The authors should provide a distribution of treatment histories for patients whose information is available. The authors should also consider re-stratifying the survival analysis for a better homogenized cohort and should determine whether this changes the key results (i.e. VAF, clonal dominance, etc.) shown throughout the paper.
2. Supp Fig 3—Given the inclusion of over 2,000 patients in this study, the authors may be powered to determine relationships of co-occurrence or mutual exclusivity with more than 2 mutations. For example, it is known that mutations in DNMT3A-NPM1-FLT3(ITD) are commonly found together with dismal prognosis, and it would be worthwhile to determine other influential mutation combinations.
3. Fig 2d and 2e— The reviewer agrees with the authors that mutations in NRAS/KRAS/PTPN11 are likely to occur in separate subclones. While these mutations all activate the RAS/MAPK signaling pathway, they have variable potencies and influence oncogenesis through distinct mechanisms of action. Moreover, mutations affecting MAPK activation are thought to be later events in leukemogenesis. These figure panels suggest that the order of mutations is determinable, but it cannot be ruled out whether cooperating mutations are influencing the VAF of these N/KRAS/PTPN11 mutations. The authors could strengthen their argument by showing the VAFs of cooperating mutations in these samples, which may give a clearer picture of clonal exclusivity/full clonal mutational patterns.
4. The authors should make it more clear how they are addressing cases in which there are multiple mutations in the same gene or pathway (i.e. RAS/MAPK-pathway mutations). Does convergent evolution influence clinical correlates or survival outcomes?
5. Fig 5— One weakness to this approach is the lack of a uniform computational pipeline. It has been shown that different mutation calling algorithms may influence clonal deconvolution using tools such as PyClone. Using a “sensitive” mutation caller may overestimate the number of clones, which in turn would influence the interpretation of whether clonal burden affects clinical outcomes. The authors are encouraged to, when available, uniformly analyze raw data and apply PyClone deconvolution to address any biases that may have been introduced in the analysis included in this study.

6. The reviewer also wonders what the mean sequencing depth is for each of the studies used for deconvolution. The authors are encouraged to comment on which criteria were applied to determine whether PyClone was appropriate? Were any samples excluded from this approach, and if so, why?

Minor Comments

1. Lines 131 and 134 – Supp Fig 4 is referenced before Supp Fig 3c.
2. Fig 3a—Why is FLT3 omitted from this panel?
3. Supp Fig 5e. – Are colors switched for “over” and “under”? In lines 174-175, the authors state that high-VAF for several mutations associated with poorer outcomes, yet the color scheme for the corresponding figure suggests the opposite.

Reviewer #3 (Remarks to the Author): Expert in AML therapy

Using a cohort of 2884 AML patients profiled by deep sequencing, the authors describe several novel features for risk stratification and prediction of sensitivity to small molecule inhibitors, showing how several features of clonal architecture including order of mutation acquisition, variant allele frequency (VAF), and branched vs linear evolutionary structure are strongly linked to clinical features and drug sensitivities.

While the evaluation of the statistical tools used to provide these data are out of the expertise of this reviewer, from a clinical/biological point of view the study raises several new concepts which are of great interest in this field and open new areas of research

Interestingly, VAFs could significantly re-stratify outcomes for 9 mutations. In addition, the authors found that the order of co-occurring NRAS and GATA2 mutations robustly stratified patient outcomes; patients where NRAS occurred before GATA2 showed remarkably poor survival compared to patients where NRAS occurred later.

The authors also rank-ordered mutations based on their relative order of acquisition and confirmed, as previously reported, that epigenetic dysregulation typically occurs early while mutations enhancing proliferation occur late in tumor development.

In this regard, the authors indicate that the ordering of mutations in functional classes significantly stratified survival. Accordingly, if a chromatin/cohesin mutation occurred before an NPM1 variant, there was a strong association with poor survival. Similarly, when transcription factor mutations occurred before splicing.

Nevertheless, this data is based on very few patients so that this conclusion should be “down-graded”.

Other than that, I have no comments and I strongly recommend to accept this interesting manuscript for publication.

We thank the reviewers for their helpful comments and present a point-by-point response below that we feel addresses each of these questions. In addition, we would like to note that 61 patients have been removed from the aggregated cohort because a server crashed and we lost access to the raw sequencing data. This resulted in 33 de novo patients being removed from several analyses in the manuscript. The discrepancy in 61 and 33 has to do with the fact that there are multiple samples (diagnosis and relapse) for some patients and that there are some de novo and secondary AML cases in the total cohort. Since all analyses were restricted to the de novo samples from diagnosis, this resulted in only 33 samples being removed from the analyses presented in the manuscript. All analyses for figures 1-4 and related supplements have been re-analyzed without this subset of patients (~1.5% of the total de novo cohort) and the main results have not changed.

Reviewer 1

We thank the reviewer for their comments that our study includes a “comprehensive analysis of sequencing data” and delivers “interesting results”.

1. *Fig 1 and Supp. Fig 1— Since this meta-analysis includes many different cohorts from different institutions, one concern that arises is that treatment history is probably heterogeneous. This is likely to influence leukemia fitness and may lead to subclonal selection in some specific contexts. The authors should provide a distribution of treatment histories for patients whose information is available. The authors should also consider re-stratifying the survival analysis for a better homogenized cohort and should determine whether this changes the key results (i.e. VAF, clonal dominance, etc.) shown throughout the paper.*

The authors agree with the reviewer that treatment heterogeneity can be a potential confounding variable when performing meta-analyses. To better understand the treatment heterogeneity present in our study, we aggregated all the patient-level data related to induction therapy, consolidation therapy, whether a transplant was performed, and what type of transplant was performed. Due to the variability in how each study reported similar treatment types and/or combinations, we decided to group induction treatments into several broad categories: (1) 7+3; (2) 7+3 + some other treatment; (3) ICE; (4) ICE + ATRA; (5) ICE + VPA + ATRA; (6) Unknown; and (7) other. For consolidation therapy, we grouped patients into the following bins: (1) Transplant; (2) HiDAC; (3) palliative; (4) targeted; (5) targeted + other; (6) none; (7) other; (8) unknown. Similarly, we collapsed transplant type into four groups: (1) Allo; (2) auto; (3) none; (4) unknown. The results from this analysis are shown below and included in Supplementary Fig. 1e-h.

Once we simplified and consolidated treatment histories, we then wanted to understand how many patients could be grouped into broadly “homogeneously-treated” bins. The results of this

analysis are shown below in the alluvial plot and now included in **Supplementary Fig. 1i**. This plot shows the flow of available treatment information for the studies included in our cohort. Ribbons are colored based on study and the width of each ribbon is scaled based on the number of patients with a subset of shared features

After counting the number of patients who fall into unique, homogeneous treatment cohorts, we found that the average group of “uniformly” treated patients was ~54 patients (SD = 99). The largest “homogeneously” treated group was a subset of 822 patients with ICE induction therapy. Using this subset of patients, we used maximally selected rank statistics to determine optimal VAF thresholds for survival stratification (as performed in **Fig. 3d**), **the results of which are shown below.**

Comparing these results with those obtained from using our aggregated cohort, we noticed minimal differences. Using our entire cohort, we had enough patients to analyze 29 genes, whereas in the “homogeneous” cohort we were only powered to analyze 25 genes. Importantly, we lost power to analyze the *GATA2* association in our subset cohort; this is unfortunate due to the *GATA2* survival associations observed in our aggregated analysis (**Fig. 3d**). Of note, seven genes (*ASXL1*, *BCOR*, *PHF6*, *NF1*, *PTPN11*, *TET2*, and *NRAS*) showed overlapping significance between our subset analysis and aggregated cohort. The only differences we noted were new survival association for *SRSF2*. Compared to our aggregated analysis, these associations

showed similar hazard ratios, but gained significance in the subset group. Together, 7/7 survival associations predicted using our aggregated cohort (and available for analysis) were confirmed using the largest “homogeneously-treated” cohort.

One of the major strengths of our work is the significantly improved power of detection for less frequent events. Additionally, many of the analyses we report are already performed on a small subset of the data. If we were to re-stratify the survival analyses based on uniformly treated patient subsets, we would unfortunately lose most of our statistical power to test the hypotheses that are central to the novelty of the body of work presented. Additionally, we would like to point out that none of the studies included in this meta-analysis (many of which are gold-standard in the field) performed their survival analyses within homogeneously treated patient populations and that this is not a common requirement in the field.

2. *Supp Fig 3—Given the inclusion of over 2,000 patients in this study, the authors may be powered to determine relationships of co-occurrence or mutual exclusivity with more than 2 mutations. For example, it is known that mutations in DNMT3A-NPM1-FLT3(ITD) are commonly found together with dismal prognosis, and it would be worthwhile to determine other influential mutation combinations.*

We agree with the reviewer that our large cohort size powers the analysis of associations related to patients with more than two mutations. To address this comment, we have analyzed the distribution of triple-mutated genotypes and their correlation with survival outcomes, the results of which are now included below and in Fig 2 d-e (lines 160-171). This analysis was worthwhile given the discovery of four novel 3-mutation genotypes with significant survival associations: $DNMT3A^{mut}FLT3-ITD^{mut}IDH1^{mut}$, $DNMT3A^{mut}FLT3-ITD^{mut}IDH2^{mut}$, and $DNMT3A^{mut}FLT3-ITD^{mut}PTPN11^{mut}$ were associated with worse outcomes while $NPM1^{mut}DNMT3A^{mut}RAD21^{mut}$ predicted good prognosis.

3. Fig 2d and 2e— The reviewer agrees with the authors that mutations in *NRAS*/*KRAS*/*PTPN11* are likely to occur in separate subclones. While these mutations all activate the *RAS*/*MAPK* signaling pathway, they have variable potencies and influence oncogenesis through distinct mechanisms of action. Moreover, mutations affecting *MAPK* activation are thought to be later events in leukemogenesis. These figure panels suggest that the order of mutations is determinable, but it cannot be ruled out whether cooperating mutations are influencing the VAF of these *N/KRAS*/*PTPN11* mutations. The authors could strengthen their argument by showing the VAFs of cooperating mutations in these samples, which may give a clearer picture of clonal exclusivity/full clonal mutational patterns.

To address this point, we have performed new analyses to show the statistically significant difference in VAF based on the assigned category of mutation order of acquisition. We have also provided the distribution of VAFs for cooperating mutations in these samples grouped by the assigned order of acquisition. When *NRAS* mutations occur before *KRAS* mutations it appears that there are more clonally dominant cooperating variants. Interestingly, when *PTPN11* mutations occur before *NRAS* mutations it appears that there are more clonally dominant cooperating variants.

4. The authors should make it more clear how they are addressing cases in which there are multiple mutations in the same gene or pathway (i.e. RAS/MAPK-pathway mutations). Does convergent evolution influence clinical correlates or survival outcomes?

The authors agree with the reviewer that there is a subset of patients with multiple mutations in the same gene (e.g., *CEBPA* or *TP53*) and that an analysis of multiple mutations in the same gene or pathway would be an interesting analysis. We have performed several new analyses to address this question and the results are shown below and now included in **Supplemental Fig. 3f-i**. At the individual gene level, there were not many significant associations with clinical features or survival, however, we have included those that were in the manuscript (lines 147-152). Compared to patients with only one mutation, multiple mutations in *CEBPA* predicted high platelet counts, lower hemoglobin counts and peripheral blood blast percentages, older age, and better survival outcomes (**Supplementary Fig. 3f-g**). Multi-hit *TP53* correlated with higher bone marrow and peripheral blood blast percentages and decreased age while multiple mutations in *FLT3-TKD* was associated with older age (**Supplementary Fig. 3f**). For the pathway/category analysis we

identified more significant associations for both clinical features and outcomes associations. Multiple mutations in genes related to transcription were associated with decreased WBC counts, hemoglobin levels, and peripheral blast percentages, while also associating with increased platelet counts and improved outcomes (**Supplementary Fig. 3h-i**). More than one mutation in tumor suppressors predicted higher bone marrow blast percentages, whereas multiple mutations in RTK/RAS signaling components correlated with improved outcomes (**Supplementary Fig. 3h-i**). Finally, multiple mutations in genes related to chromatin remodeling and cohesion components correlated with lower WBC counts, younger age, and worse survival outcomes (**Supplementary Fig. 3h-i**). Description of these associations are now also included in the manuscript (lines 152-159).

5. *Fig 5— One weakness to this approach is the lack of a uniform computational pipeline. It has been shown that different mutation calling algorithms may influence clonal deconvolution using tools such as PyClone. Using a “sensitive” mutation caller may overestimate the number of clones, which in turn would influence the interpretation of whether clonal burden affects clinical outcomes. The authors are encouraged to, when available, uniformly analyze raw data and apply PyClone deconvolution to address any biases that may have been introduced in the analysis included in this study.*

We agree with the reviewer that differences in mutation callers can introduce significant differences in the detection of mutations and confounding effects in the resulting downstream analyses. Fortunately, both studies we analyzed with PyClone (TCGA and Beat AML) used VarScan2 (a gold-standard variant caller) for their variant calling. Although the Beat AML study used VarScan 2 and MuTect for variant detection, we only used the consensus call from VarScan 2 and MuTect for PyClone analysis. Therefore, we believe that the raw data has already been uniformly analyzed and that any further reprocessing of the raw data is unnecessary.

However, we agree that a more in-depth analysis of potential differences between these cohorts is worth addressing. As such, we have performed several new analyses to show that there are minimal differences in the input features used by PyClone based on the two cohorts. We first looked at the VAF distribution for the most frequent mutations used for PyClone analysis. This analysis was intended to address the concern that there might be cohort-level differences in the clonal/subclonal distribution of mutations used for clonal modeling. For all but two mutations (*TET2* and *FLT3-TKD*), there was no statistical difference between VAF distributions for the two cohorts. This indicates that there is not a significant bias in the input data used by PyClone.

We also tested whether PyClone results showed major skews in the number of clones predicted based on cohort. The primary question we sought to answer was whether there might be biases in the number of clones predicted by PyClone and whether this was associated with underlying differences in mutation VAFs between the two cohorts. As shown below, almost all clone bins show no statistical difference in VAF distribution between cohorts. Additionally, for 2/3 showing significant differences ($n = 6$ and $n = 8$), the effect size between cohorts is very small and not likely biologically relevant.

6. The reviewer also wonders what the mean sequencing depth is for each of the studies used for deconvolution. The authors are encouraged to comment on which criteria were applied to determine whether PyClone was appropriate? Were any samples excluded from this approach, and if so, why?

Mean sequencing depth for the Beat AML study was 277x and for TCGA was 132x. The authors believe that this is sufficient sequencing depth to perform PyClone analysis, especially given that a recent iteration of PyClone from the original creators uses a mean sequencing depth of 100x¹. Several samples (<10) were excluded from the analysis due to insufficient number of driver mutations (n = 0) or a failure to be processed by the PyClone pipeline due to technical limitations of memory allocation in the PyClone algorithm.

1. Gillis, S., Roth, A. PyClone-VI: scalable inference of clonal population structures using whole genome data. *BMC Bioinformatics* **21**, 571 (2020).

Minor Comments:

1. Lines 131 and 134 – Supp Fig 4 is referenced before Supp Fig 3c.

Supplemental figure 4 has been replaced and the order of reference has been fixed.

2. Fig 3a—Why is FLT3 omitted from this panel?

This omission was unintentional and FLT3-ITD and FLT3-TKD VAF distributions have now been added to the panel.

3. Supp Fig 5e. – Are colors switched for “over” and “under”? In lines 174-175, the authors state that high-VAF for several mutations associated with poorer outcomes, yet the color scheme for the corresponding figure suggests the opposite.

The reviewer is correct – the colors were accidentally switched for “over” and “under” in this panel. This issue has been addressed. Thanks for your careful review.

e

Reviewer 3

We thank the reviewer for their comment that “I strongly recommend to accept this interesting manuscript”.

1. *In this regard, the authors indicate that the ordering of mutations in functional classes significantly stratified survival. Accordingly, if a chromatin/cohesin mutation occurred before an NPM1 variant, there was a strong association with poor survival. Similarly, when transcription factor mutations occurred before splicing. Nevertheless, this data is based on very few patients so that this conclusion should be “down-graded”.*

We agree with the reviewer that these survival associations are based on relatively small groups of patients and that this fact should be noted more clearly in the manuscript. To draw attention to this point, we have now included the following statement addressing this in lines 341-343, 346-347:

“However, even with our cohort size, analyzing survival by mutation order remains weakly powered for less frequent genotypes and survival differences based on the ordering of mutation categories represent relatively few patients.... Future studies in even larger cohorts will be needed to address this question more precisely.”

REVIEWERS' COMMENTS

Reviewer #1 (Remarks to the Author):

The reviewer thanks the authors for their detailed responses to the questions/comments raised during the initial revision of the manuscript. Since re-stratifying the cohort based on treatment history showed a minimal influence over the results, the reviewer believes it is sufficient to use the aggregate analysis as it was presented in the initial submission of the manuscript. It is the reviewer's opinion that the revised analyses conducted by the authors strengthen the manuscript and should be included in the final version. The reviewer believes that the manuscript is interesting, timely, and should be accepted for publication.

Reviewer #3 (Remarks to the Author):

The authors have properly answered reviewer comment

We thank the reviewers for their time in addressing our response to their comments and for recommending our work for publication.

Reviewer #1 (Remarks to the Author):

The reviewer thanks the authors for their detailed responses to the questions/comments raised during the initial revision of the manuscript. Since re-stratifying the cohort based on treatment history showed a minimal influence over the results, the reviewer believes it is sufficient to use the aggregate analysis as it was presented in the initial submission of the manuscript. It is the reviewer's opinion that the revised analyses conducted by the authors strengthen the manuscript and should be included in the final version. The reviewer believes that the manuscript is interesting, timely, and should be accepted for publication.

We agree with the reviewer that the minimal differences in survival after stratifying based on uniform treatment history gives more confidence to the aggregated analysis. We also agree that the revised analyses suggested by the reviewer improve the manuscript and we will incorporate them into the final version. We thank the reviewer for finding our manuscript interesting and timely and for recommending it for publication.

Reviewer #3 (Remarks to the Author):

The authors have properly answered reviewer comment

We thank the reviewer for agreeing that we have sufficiently answered their comments.